# The Pareto Regret Frontier for Bandits

**Tor Lattimore**
Department of Computing Science
University of Alberta, Canada
`tor.lattimore@gmail.com`

## Abstract

Given a multi-armed bandit problem it may be desirable to achieve a smaller-than-usual worst-case regret for some special actions. I show that the price for such unbalanced worst-case regret guarantees is rather high. Specifically, if an algorithm enjoys a worst-case regret of $B$ with respect to some action, then there must exist another action for which the worst-case regret is at least $\Omega(nK/B)$, where $n$ is the horizon and $K$ the number of actions. I also give upper bounds in both the stochastic and adversarial settings showing that this result cannot be improved. For the stochastic case the pareto regret frontier is characterised exactly up to constant factors.

## 1 Introduction

The multi-armed bandit is the simplest class of problems that exhibit the exploration/exploitation dilemma. In each time step the learner chooses one of $K$ actions and receives a noisy reward signal for the chosen action. A learner's performance is measured in terms of the regret, which is the (expected) difference between the rewards it actually received and those it would have received (in expectation) by choosing the optimal action.

Prior work on the regret criterion for finite-armed bandits has treated all actions uniformly and has aimed for bounds on the regret that do not depend on which action turned out to be optimal. I take a different approach and ask what can be achieved if some actions are given special treatment. Focussing on worst-case bounds, I ask whether or not it is possible to achieve improved worst-case regret for some actions, and what is the cost in terms of the regret for the remaining actions. Such results may be useful in a variety of cases. For example, a company that is exploring some new strategies might expect an especially small regret if its existing strategy turns out to be (nearly) optimal.

This problem has previously been considered in the experts setting where the learner is allowed to observe the reward for all actions in every round, not only for the action actually chosen. The earliest work seems to be by Hutter and Poland [2005] where it is shown that the learner can assign a prior weight to each action and pays a worst-case regret of $O(\sqrt{-n\log\rho_i})$ for expert $i$ where $\rho_i$ is the prior belief in expert $i$ and $n$ is the horizon. The uniform regret is obtained by choosing $\rho_i = 1/K$, which leads to the well-known $O(\sqrt{n\log K})$ bound achieved by the exponential weighting algorithm [Cesa-Bianchi, 2006]. The consequence of this is that an algorithm can enjoy a constant regret with respect to a single action while suffering minimally on the remainder. The problem was studied in more detail by Koolen [2013] where (remarkably) the author was able to *exactly* describe the pareto regret frontier when $K = 2$.

Other related work (also in the experts setting) is where the objective is to obtain an improved regret against a mixture of available experts/actions [Even-Dar et al., 2008, Kapralov and Panigrahy, 2011]. In a similar vain, Sani et al. [2014] showed that algorithms for prediction with expert advice can be combined with minimal cost to obtain the best of both worlds. In the bandit setting I am only aware

of the work by Liu and Li [2015] who study the effect of the prior on the regret of Thompson sampling in a special case. In contrast the lower bound given here applies to all algorithms in a relatively standard setting.

The main contribution of this work is a characterisation of the pareto regret frontier (the set of achievable worst-case regret bounds) for stochastic bandits.

Let $\mu_i \in \mathbb{R}$ be the unknown mean of the $i$th arm and assume that $\sup_{i,j} \mu_i - \mu_j \leq 1$. In each time step the learner chooses an action $I_t \in \{1, \ldots, K\}$ and receives reward $g_{I_t,t} = \mu_i + \eta_t$ where $\eta_t$ is the noise term that I assume to be sampled independently from a 1-subgaussian distribution that may depend on $I_t$. This model subsumes both Gaussian and Bernoulli (or bounded) rewards. Let $\pi$ be a bandit strategy, which is a function from histories of observations to an action $I_t$. Then the $n$-step expected pseudo regret with respect to the $i$th arm is

$$R_{\mu,i}^{\pi} = n\mu_i - \mathbb{E} \sum_{t=1}^{n} \mu_{I_t} \, ,$$

where the expectation is taken with respect to the randomness in the noise and the actions of the policy. Throughout this work $n$ will be fixed, so is omitted from the notation. The worst-case expected pseudo-regret with respect to arm $i$ is

$$R_i^{\pi} = \sup_{\mu} R_{\mu,i}^{\pi} \, . \tag{1}$$

This means that $R^{\pi} \in \mathbb{R}^K$ is a vector of worst-case pseudo regrets with respect to each of the arms. Let $\mathcal{B} \subset \mathbb{R}^K$ be a set defined by

$$\mathcal{B} = \left\{ B \in [0,n]^K : B_i \geq \min \left\{ n, \sum_{j \neq i} \frac{n}{B_j} \right\} \text{ for all } i \right\} \, . \tag{2}$$

The boundary of $\mathcal{B}$ is denoted by $\delta\mathcal{B}$. The following theorem shows that $\delta\mathcal{B}$ describes the pareto regret frontier up to constant factors.

---

**Theorem**

There exist universal constants $c_1 = 8$ and $c_2 = 252$ such that:

**Lower bound:** for $\eta_t \sim \mathcal{N}(0,1)$ and all strategies $\pi$ we have $c_1(R^{\pi} + K) \in \mathcal{B}$

**Upper bound:** for all $B \in \mathcal{B}$ there exists a strategy $\pi$ such that $R_i^{\pi} \leq c_2 B_i$ for all $i$

---

Observe that the lower bound relies on the assumption that the noise term be Gaussian while the upper bound holds for subgaussian noise. The lower bound may be generalised to other noise models such as Bernoulli, but does not hold for all subgaussian noise models. For example, it does not hold if there is no noise ($\eta_t = 0$ almost surely).

The lower bound also applies to the adversarial framework where the rewards may be chosen arbitrarily. Although I was not able to derive a matching upper bound in this case, a simple modification of the Exp-$\gamma$ algorithm [Bubeck and Cesa-Bianchi, 2012] leads to an algorithm with

$$R_1^{\pi} \leq B_1 \quad \text{and} \quad R_k^{\pi} \lesssim \frac{nK}{B_1} \log\left(\frac{nK}{B_1^2}\right) \text{ for all } k \geq 2 \, ,$$

where the regret is the adversarial version of the expected regret. Details are in the supplementary material.

The new results seem elegant, but disappointing. In the experts setting we have seen that the learner can distribute a prior amongst the actions and obtain a bound on the regret depending in a natural way on the prior weight of the optimal action. In contrast, in the bandit setting the learner pays an enormously higher price to obtain a small regret with respect to even a single arm. In fact, the learner must essentially choose a single arm to favour, after which the regret for the remaining arms has very limited flexibility. Unlike in the experts setting, if even a single arm enjoys constant worst-case regret, then the worst-case regret with respect to all other arms is necessarily linear.

## 2 Preliminaries

I use the same notation as Bubeck and Cesa-Bianchi [2012]. Define $T_i(t)$ to be the number of times action $i$ has been chosen after time step $t$ and $\hat{\mu}_{i,s}$ to be the empirical estimate of $\mu_i$ from the first $s$ times action $i$ was sampled. This means that $\hat{\mu}_{i,T_i(t-1)}$ is the empirical estimate of $\mu_i$ at the start of the $t$th round. I use the convention that $\hat{\mu}_{i,0} = 0$. Since the noise model is 1-subgaussian we have

$$\forall \varepsilon > 0 \qquad \mathbb{P}\left\{\exists s \leq t : \hat{\mu}_{i,s} - \mu_i \geq \varepsilon/s\right\} \leq \exp\left(-\frac{\varepsilon^2}{2t}\right). \tag{3}$$

This result is presumably well known, but a proof is included in the supplementary material for convenience. The optimal arm is $i^* = \arg\max_i \mu_i$ with ties broken in some arbitrary way. The optimal reward is $\mu^* = \max_i \mu_i$. The gap between the mean rewards of the $j$th arm and the optimal arm is $\Delta_j = \mu^* - \mu_j$ and $\Delta_{ji} = \mu_i - \mu_j$. The vector of worst-case regrets is $R^\pi \in \mathbb{R}^K$ and has been defined already in Eq. (1). I write $R^\pi \leq B \in \mathbb{R}^K$ if $R_i^\pi \leq B_i$ for all $i \in \{1, \ldots, K\}$. For vector $R^\pi$ and $x \in \mathbb{R}$ we have $(R^\pi + x)_i = R_i^\pi + x$.

## 3 Understanding the Frontier

Before proving the main theorem I briefly describe the features of the regret frontier. First notice that if $B_i = \sqrt{n(K-1)}$ for all $i$, then

$$B_i = \sqrt{n(K-1)} = \sum_{j \neq i} \sqrt{n/(K-1)} = \sum_{j \neq i} \frac{n}{B_j}.$$

Thus $B \in \mathcal{B}$ as expected. This particular $B$ is witnessed up to constant factors by MOSS [Audibert and Bubeck, 2009] and OC-UCB [Lattimore, 2015], but not UCB [Auer et al., 2002], which suffers $R_i^{\text{ucb}} \in \Omega(\sqrt{nK\log n})$.

Of course the uniform choice of $B$ is not the only option. Suppose the first arm is special, so $B_1$ should be chosen especially small. Assume without loss of generality that $B_1 \leq B_2 \leq \ldots \leq B_K \leq n$. Then by the main theorem we have

$$B_1 \geq \sum_{i=2}^{K} \frac{n}{B_i} \geq \sum_{i=2}^{k} \frac{n}{B_i} \geq \frac{(k-1)n}{B_k}.$$

Therefore

$$B_k \geq \frac{(k-1)n}{B_1}. \tag{4}$$

This also proves the claim in the abstract, since it implies that $B_K \geq (K-1)n/B_1$. If $B_1$ is fixed, then choosing $B_k = (k-1)n/B_1$ does not lie on the frontier because

$$\sum_{k=2}^{K} \frac{n}{B_k} = \sum_{k=2}^{K} \frac{B_1}{k-1} \in \Omega(B_1 \log K)$$

However, if $H = \sum_{k=2}^{K} 1/(k-1) \in \Theta(\log K)$, then choosing $B_k = (k-1)nH/B_1$ does lie on the frontier and is a factor of $\log K$ away from the lower bound given in Eq. (4). Therefore up the a $\log K$ factor, points on the regret frontier are characterised entirely by a permutation determining the order of worst-case regrets and the smallest worst-case regret.

Perhaps the most natural choice of $B$ (assuming again that $B_1 \leq \ldots \leq B_K$) is

$$B_1 = n^p \qquad \text{and} \qquad B_k = (k-1)n^{1-p}H \text{ for } k > 1.$$

For $p = 1/2$ this leads to a bound that is at most $\sqrt{K}\log K$ worse than that obtained by MOSS and OC-UCB while being a factor of $\sqrt{K}$ better for a select few.

**Assumptions**

The assumption that $\Delta_i \in [0,1]$ is used to avoid annoying boundary problems caused by the fact that time is discrete. This means that if $\Delta_i$ is extremely large, then even a single sample from this arm can cause a big regret bound. This assumption is already quite common, for example a worst-case regret of $\Omega(\sqrt{Kn})$ clearly does not hold if the gaps are permitted to be unbounded. Unfortunately there is no perfect resolution to this annoyance. Most elegant would be to allow time to be continuous with actions taken up to stopping times. Otherwise you have to deal with the discretisation/boundary problem with special cases, or make assumptions as I have done here.

## 4 Lower Bounds

**Theorem 1.** *Assume $\eta_t \sim \mathcal{N}(0,1)$ is sampled from a standard Gaussian. Let $\pi$ be an arbitrary strategy, then $8(R^\pi + K) \in \mathcal{B}$.*

*Proof.* Assume without loss of generality that $R_1^\pi = \min_i R_i^\pi$ (if this is not the case, then simply re-order the actions). If $R_1^\pi > n/8$, then the result is trivial. From now on assume $R_1^\pi \le n/8$. Let $c = 4$ and define

$$\varepsilon_k = \min\left\{\frac{1}{2}, \frac{cR_k^\pi}{n}\right\} \le \frac{1}{2}.$$

Define $K$ vectors $\mu_1, \ldots, \mu_K \in \mathbb{R}^K$ by

$$(\mu_k)_j = \frac{1}{2} + \begin{cases} 0 & \text{if } j = 1 \\ \varepsilon_k & \text{if } j = k \ne 1 \\ -\varepsilon_j & \text{otherwise}. \end{cases}$$

Therefore the optimal action for the bandit with means $\mu_k$ is $k$. Let $A = \{k : R_k^\pi \le n/8\}$ and $A' = \{k : k \notin A\}$ and assume $k \in A$. Then

$$R_k^\pi \overset{(a)}{\ge} R_{\mu_k,k}^\pi \overset{(b)}{\ge} \varepsilon_k \mathbb{E}_{\mu_k}^\pi \left[\sum_{j \ne k} T_j(n)\right] \overset{(c)}{=} \varepsilon_k \left(n - \mathbb{E}_{\mu_k}^\pi T_k(n)\right) \overset{(d)}{=} \frac{cR_k^\pi(n - \mathbb{E}_{\mu_k}^\pi T_k(n))}{n},$$

where (a) follows since $R_k^\pi$ is the worst-case regret with respect to arm $k$, (b) since the gap between the means of the $k$th arm and any other arm is at least $\varepsilon_k$ (Note that this is also true for $k = 1$ since $\varepsilon_1 = \min_k \varepsilon_k$. (c) follows from the fact that $\sum_i T_i(n) = n$ and (d) from the definition of $\varepsilon_k$. Therefore

$$n\left(1 - \frac{1}{c}\right) \le \mathbb{E}_{\mu_k}^\pi T_k(n). \tag{5}$$

Therefore for $k \ne 1$ with $k \in A$ we have

$$n\left(1 - \frac{1}{c}\right) \le \mathbb{E}_{\mu_k}^\pi T_k(n) \overset{(a)}{\le} \mathbb{E}_{\mu_1}^\pi T_k(n) + n\varepsilon_k \sqrt{\mathbb{E}_{\mu_1}^\pi T_k(n)}$$

$$\overset{(b)}{\le} n - \mathbb{E}_{\mu_1}^\pi T_1(n) + n\varepsilon_k \sqrt{\mathbb{E}_{\mu_1}^\pi T_k(n)} \overset{(c)}{\le} \frac{n}{c} + n\varepsilon_k \sqrt{\mathbb{E}_{\mu_1}^\pi T_k(n)},$$

where (a) follows from standard entropy inequalities and a similar argument as used by Auer et al. [1995] (details in supplementary material), (b) since $k \ne 1$ and $\mathbb{E}_{\mu_1}^\pi T_1(n) + \mathbb{E}_{\mu_1}^\pi T_k(n) \le n$, and (c) by Eq. (5). Therefore

$$\mathbb{E}_{\mu_1}^\pi T_k(n) \ge \frac{1 - \frac{2}{c}}{\varepsilon_k^2},$$

which implies that

$$R_1^\pi \ge R_{\mu_1,1}^\pi = \sum_{k=2}^K \varepsilon_k \mathbb{E}_{\mu_1}^\pi T_k(n) \ge \sum_{k \in A - \{1\}} \frac{1 - \frac{2}{c}}{\varepsilon_k} = \frac{1}{8} \sum_{k \in A - \{1\}} \frac{n}{R_k^\pi}.$$

Therefore for all $i \in A$ we have

$$8R_i^\pi \geq \sum_{k \in A - \{1\}} \frac{n}{R_k^\pi} \cdot \frac{R_i^\pi}{R_1^\pi} \geq \sum_{k \in A - \{i\}} \frac{n}{R_k^\pi} \, .$$

Therefore

$$8R_i^\pi + 8K \geq \sum_{k \neq i} \frac{n}{R_k^\pi} + 8K - \sum_{k \in A' - \{i\}} \frac{n}{R_k^\pi} \geq \sum_{k \neq i} \frac{n}{R_k^\pi} \, ,$$

which implies that $8(R^\pi + K) \in \mathcal{B}$ as required. $\qquad\square$

## 5   Upper Bounds

I now show that the lower bound derived in the previous section is tight up to constant factors. The algorithm is a generalisation MOSS [Audibert and Bubeck, 2009] with two modifications. First, the width of the confidence bounds are biased in a non-uniform way, and second, the upper confidence bounds are shifted. The new algorithm is functionally identical to MOSS in the special case that $B_i$ is uniform. Define $\log_+(x) = \max\{0, \log(x)\}$.

---

1: **Input:** $n$ and $B_1, \ldots, B_K$
2: $n_i = n^2 / B_i^2$ for all $i$
3: **for** $t \in 1, \ldots, n$ **do**

4: $\qquad I_t = \arg\max_i \hat{\mu}_{i, T_i(t-1)} + \sqrt{\frac{4}{T_i(t-1)} \log_+\left(\frac{n_i}{T_i(t-1)}\right)} - \sqrt{\frac{1}{n_i}}$

5: **end for**

---

**Algorithm 1:** Unbalanced MOSS

**Theorem 2.** *Let $B \in \mathcal{B}$, then the strategy $\pi$ given in Algorithm 1 satisfies $R^\pi \leq 252B$.*

**Corollary 3.** *For all $\mu$ the following hold:*

1. $R_{\mu, i^*}^\pi \leq 252 B_{i^*}$.

2. $R_{\mu, i^*}^\pi \leq \min_i (n\Delta_i + 252 B_i)$

The second part of the corollary is useful when $B_{i^*}$ is large, but there exists an arm for which $n\Delta_i$ and $B_i$ are both small. The proof of Theorem 2 requires a few lemmas. The first is a somewhat standard concentration inequality that follows from a combination of the peeling argument and Doob's maximal inequality. The proof may be found in the supplementary material.

**Lemma 4.** *Let $Z_i = \max_{1 \leq s \leq n} \mu_i - \hat{\mu}_{i,s} - \sqrt{\frac{4}{s} \log_+\left(\frac{n_i}{s}\right)}$. Then $\mathbb{P}\{Z_i \geq \Delta\} \leq \frac{20}{n_i \Delta^2}$ for all $\Delta > 0$.*

In the analysis of traditional bandit algorithms the gap $\Delta_{ji}$ measures how quickly the algorithm can detect the difference between arms $i$ and $j$. By design, however, Algorithm 1 is negatively biasing its estimate of the empirical mean of arm $i$ by $\sqrt{1/n_i}$. This has the effect of shifting the gaps, which I denote by $\bar{\Delta}_{ji}$ and define to be

$$\bar{\Delta}_{ji} = \Delta_{ji} + \sqrt{1/n_j} - \sqrt{1/n_i} = \mu_i - \mu_j + \sqrt{1/n_j} - \sqrt{1/n_i} \, .$$

**Lemma 5.** *Define stopping time $\tau_{ji}$ by*

$$\tau_{ji} = \min\left\{ s : \hat{\mu}_{j,s} + \sqrt{\frac{4}{s} \log_+\left(\frac{n_j}{s}\right)} \leq \mu_j + \bar{\Delta}_{ji}/2 \right\} \, .$$

*If $Z_i < \bar{\Delta}_{ji}/2$, then $T_j(n) \leq \tau_{ji}$.*

*Proof.* Let $t$ be the first time step such that $T_j(t-1) = \tau_{ji}$. Then

$$\hat{\mu}_{j,T_j(t-1)} + \sqrt{\frac{4}{T_j(t-1)} \log_+\left(\frac{n_j}{T_j(t-1)}\right)} - \sqrt{1/n_j} \leq \mu_j + \bar{\Delta}_{ji}/2 - \sqrt{1/n_j}$$

$$= \mu_j + \bar{\Delta}_{ji} - \bar{\Delta}_{ji}/2 - \sqrt{1/n_j}$$

$$= \mu_i - \sqrt{1/n_i} - \bar{\Delta}_{ji}/2$$

$$< \hat{\mu}_{i,T_i(t-1)} + \sqrt{\frac{4}{T_i(t-1)} \log_+\left(\frac{n_i}{T_i(t-1)}\right)} - \sqrt{1/n_i} \,,$$

which implies that arm $j$ will not be chosen at time step $t$ and so also not for any subsequent time steps by the same argument and induction. Therefore $T_j(n) \leq \tau_{ji}$. $\qquad\square$

**Lemma 6.** *If $\bar{\Delta}_{ji} > 0$, then $\mathbb{E}\tau_{ji} \leq \dfrac{40}{\bar{\Delta}_{ji}^2} + \dfrac{64}{\bar{\Delta}_{ji}^2} \operatorname{ProductLog}\left(\dfrac{n_j \bar{\Delta}_{ji}^2}{64}\right).$*

*Proof.* Let $s_0$ be defined by

$$s_0 = \left\lceil \frac{64}{\bar{\Delta}_{ji}^2} \operatorname{ProductLog}\left(\frac{n_j \bar{\Delta}_{ji}^2}{64}\right) \right\rceil \quad \implies \quad \sqrt{\frac{4}{s_0} \log_+\left(\frac{n_j}{s_0}\right)} \leq \frac{\bar{\Delta}_{ji}}{4} \,.$$

Therefore

$$\mathbb{E}\tau_{ji} = \sum_{s=1}^{n} \mathbb{P}\{\tau_{ji} \geq s\} \leq 1 + \sum_{s=1}^{n-1} \mathbb{P}\left\{\hat{\mu}_{i,s} - \mu_{i,s} \geq \frac{\bar{\Delta}_{ji}}{2} - \sqrt{\frac{4}{s} \log_+\left(\frac{n_j}{s}\right)}\right\}$$

$$\leq 1 + s_0 + \sum_{s=s_0+1}^{n-1} \mathbb{P}\left\{\hat{\mu}_{i,s} - \mu_{i,s} \geq \frac{\bar{\Delta}_{ji}}{4}\right\} \leq 1 + s_0 + \sum_{s=s_0+1}^{\infty} \exp\left(-\frac{s\bar{\Delta}_{ji}^2}{32}\right)$$

$$\leq 1 + s_0 + \frac{32}{\bar{\Delta}_{ji}^2} \leq \frac{40}{\bar{\Delta}_{ji}^2} + \frac{64}{\bar{\Delta}_{ji}^2} \operatorname{ProductLog}\left(\frac{n_j \bar{\Delta}_{ji}^2}{64}\right) \,,$$

where the last inequality follows since $\bar{\Delta}_{ji} \leq 2$. $\qquad\square$

*Proof of Theorem 2.* Let $\Delta = 2/\sqrt{n_i}$ and $A = \{j : \Delta_{ji} > \Delta\}$. Then for $j \in A$ we have $\Delta_{ji} \leq 2\bar{\Delta}_{ji}$ and $\bar{\Delta}_{ji} \geq \sqrt{1/n_i} + \sqrt{1/n_j}$. Letting $\Delta' = \sqrt{1/n_i}$ we have

$$R_{\mu,i}^{\pi} = \mathbb{E}\left[\sum_{j=1}^{K} \Delta_{ji} T_j(n)\right]$$

$$\leq n\Delta + \mathbb{E}\left[\sum_{j \in A} \Delta_{ji} T_j(n)\right]$$

$$\overset{(a)}{\leq} 2B_i + \mathbb{E}\left[\sum_{j \in A} \Delta_{ji} \tau_{ji} + n \max_{j \in A}\{\Delta_{ji} : Z_i \geq \bar{\Delta}_{ji}/2\}\right]$$

$$\overset{(b)}{\leq} 2B_i + \sum_{j \in A}\left(\frac{80}{\bar{\Delta}_{ji}} + \frac{128}{\bar{\Delta}_{ji}} \operatorname{ProductLog}\left(\frac{n_j \bar{\Delta}_{ji}^2}{64}\right)\right) + 4n\mathbb{E}[Z_i \mathbb{1}\{Z_i \geq \Delta'\}]$$

$$\overset{(c)}{\leq} 2B_i + \sum_{j \in A} 90\sqrt{n_j} + 4n\mathbb{E}[Z_i \mathbb{1}\{Z_i \geq \Delta'\}] \,,$$

where (a) follows by using Lemma 5 to bound $T_j(n) \leq \tau_{ji}$ when $Z_i < \bar{\Delta}_{ji}$. On the other hand, the total number of pulls for arms $j$ for which $Z_i \geq \bar{\Delta}_{ji}/2$ is at most $n$. (b) follows by bounding

$\tau_{ji}$ in expectation using Lemma 6. (c) follows from basic calculus and because for $j \in A$ we have $\bar{\Delta}_{ji} \geq \sqrt{1/n_i}$. All that remains is to bound the expectation.

$$4n\mathbb{E}[Z_i\mathbb{1}\{Z_i \geq \Delta'\}] \leq 4n\Delta'\mathbb{P}\{Z_i \geq \Delta'\} + 4n\int_{\Delta'}^{\infty}\mathbb{P}\{Z_i \geq z\}\,dz \leq \frac{160n}{\Delta'n_i} = \frac{160n}{\sqrt{n_i}} = 160B_i\,,$$

where I have used Lemma 4 and simple identities. Putting it together we obtain

$$R_{\mu,i}^{\pi} \leq 2B_i + \sum_{j \in A} 90\sqrt{n_j} + 160B_1 \leq 252B_i\,,$$

where I applied the assumption $B \in \mathcal{B}$ and so $\sum_{j \neq 1}\sqrt{n_j} = \sum_{j \neq 1} n/B_j \leq B_i$. $\qquad\square$

The above proof may be simplified in the special case that $B$ is uniform where we recover the minimax regret of MOSS, but with perhaps a simpler proof than was given originally by Audibert and Bubeck [2009].

**On Logarithmic Regret**

In a recent technical report I demonstrated empirically that MOSS suffers sub-optimal problem-dependent regret in terms of the minimum gap [Lattimore, 2015]. Specifically, it can happen that

$$R_{\mu,i^*}^{\text{moss}} \in \Omega\left(\frac{K}{\Delta_{\min}}\log n\right)\,, \tag{6}$$

where $\Delta_{\min} = \min_{i:\Delta_i > 0}\Delta_i$. On the other hand, the order-optimal asymptotic regret can be significantly smaller. Specifically, UCB by Auer et al. [2002] satisfies

$$R_{\mu,i^*}^{\text{ucb}} \in O\left(\sum_{i:\Delta_i > 0}\frac{1}{\Delta_i}\log n\right)\,, \tag{7}$$

which for unequal gaps can be much smaller than Eq. (6) and is asymptotically order-optimal [Lai and Robbins, 1985]. The problem is that MOSS explores only enough to obtain minimax regret, but sometimes obtains minimax regret even when a more conservative algorithm would do better. It is worth remarking that this effect is harder to observe than one might think. The example given in the afforementioned technical report is carefully tuned to exploit this failing, but still requires $n = 10^9$ and $K = 10^3$ before significant problems arise. In all other experiments MOSS was performing admirably in comparison to UCB.

All these problems can be avoided by modifying UCB rather than MOSS. The cost is a factor of $O(\sqrt{\log n})$. The algorithm is similar to Algorithm 1, but chooses the action that maximises the following index.

$$I_t = \arg\max_i \hat{\mu}_{i,T_i(t-1)} + \sqrt{\frac{(2+\varepsilon)\log t}{T_i(t-1)}} - \sqrt{\frac{\log n}{n_i}}\,,$$

where $\varepsilon > 0$ is a fixed arbitrary constant.

**Theorem 7.** *If $\pi$ is the strategy of unbalanced UCB with $n_i = n^2/B_i^2$ and $B \in \mathcal{B}$, then the regret of the unbalanced UCB satisfies:*

1. *(problem-independent regret).* $R_{\mu,i^*}^{\pi} \in O\left(B_{i^*}\sqrt{\log n}\right)$.

2. *(problem-dependent regret). Let $A = \left\{i : \Delta_i \geq 2\sqrt{1/n_{i^*}\log n}\right\}$. Then*

$$R_{\mu,i^*}^{\pi} \in O\left(B_{i^*}\sqrt{\log n}\mathbb{1}\{A \neq \emptyset\} + \sum_{i \in A}\frac{1}{\Delta_i}\log n\right)\,.$$

The proof is deferred to the supplementary material. The indicator function in the problem-dependent bound vanishes for sufficiently large $n$ provided $n_{i^*} \in \omega(\log(n))$, which is equivalent to

$B_{i^*} \in o(n/\sqrt{\log n})$. Thus for reasonable choices of $B_1, \ldots, B_K$ the algorithm is going to enjoy the same asymptotic performance as UCB. Theorem 7 may be proven for any index-based algorithm for which it can be shown that

$$\mathbb{E}T_i(n) \in O\left(\frac{1}{\Delta_i^2} \log n\right),$$

which includes (for example) KL-UCB [Cappé et al., 2013] and Thompson sampling (see analysis by Agrawal and Goyal [2012a,b] and original paper by Thompson [1933]), but not OC-UCB [Lattimore, 2015] or MOSS [Audibert and Bubeck, 2009].

**Experimental Results**

I compare MOSS and unbalanced MOSS in two simple simulated examples, both with horizon $n = 5000$. Each data point is an empirical average of $\sim 10^4$ i.i.d. samples, so error bars are too small to see. Code/data is available in the supplementary material. The first experiment has $K = 2$ arms and $B_1 = n^{\frac{1}{3}}$ and $B_2 = n^{\frac{2}{3}}$. I plotted the results for $\mu = (0, -\Delta)$ for varying $\Delta$. As predicted, the new algorithm performs significantly better than MOSS for positive $\Delta$, and significantly worse otherwise (Fig. 1). The second experiment has $K = 10$ arms. This time $B_1 = \sqrt{n}$ and $B_k = (k-1)H\sqrt{n}$ with $H = \sum_{k=1}^{9} 1/k$. Results are shown for $\mu_k = \Delta \mathbb{1}\{k = i^*\}$ for $\Delta \in [0, 1/2]$ and $i^* \in \{1, \ldots, 10\}$. Again, the results agree with the theory. The unbalanced algorithm is superior to MOSS for $i^* \in \{1, 2\}$ and inferior otherwise (Fig. 2).

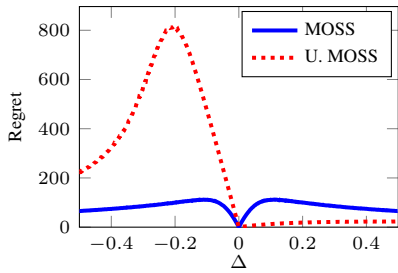

**Figure 1**

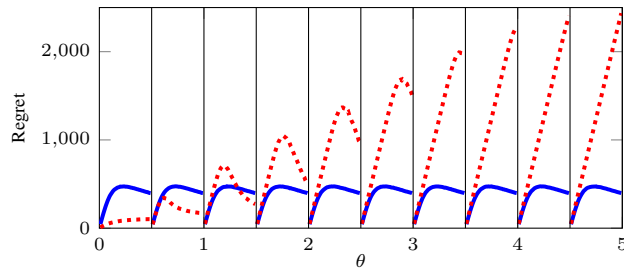

**Figure 2:** $\theta = \Delta + (i^* - 1)/2$

Sadly the experiments serve only to highlight the plight of the biased learner, which suffers significantly worse results than its unbaised counterpart for most actions.

## 6 Discussion

I have shown that the cost of favouritism for multi-armed bandit algorithms is rather serious. If an algorithm exhibits a small worst-case regret for a specific action, then the worst-case regret of the remaining actions is necessarily significantly larger than the well-known uniform worst-case bound of $\Omega(\sqrt{Kn})$. This unfortunate result is in stark contrast to the experts setting for which there exist algorithms that suffer constant regret with respect to a single expert at almost no cost for the remainder. Surprisingly, the best achievable (non-uniform) worst-case bounds are determined up to a permutation almost entirely by the value of the smallest worst-case regret.

There are some interesting open questions. Most notably, in the adversarial setting I am not sure if the upper or lower bound is tight (or neither). It would also be nice to know if the constant factors can be determined exactly asymptotically, but so far this has not been done even in the uniform case. For the stochastic setting it is natural to ask if the OC-UCB algorithm can also be modified. Intuitively one would expect this to be possible, but it would require re-working the very long proof.

**Acknowledgements**

I am indebted to the very careful reviewers who made many suggestions for improving this paper. Thank you!

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
