[Supplementary Material]

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

*Proof.* Using the peeling device.

$$\mathbb{P}\{Z_i \geq \Delta\} \overset{(a)}{=} \mathbb{P}\left\{\exists s \leq n : \mu_i - \hat{\mu}_{i,s} \geq \Delta + \sqrt{\frac{4}{s} \log_+\left(\frac{n_i}{s}\right)}\right\}$$

$$\overset{(b)}{\leq} \sum_{k=0}^{\infty} \mathbb{P}\left\{\exists s < 2^{k+1} : s(\mu_i - \hat{\mu}_{i,s}) \geq 2^k \Delta + \sqrt{2^{k+2} \log_+\left(\frac{n_i}{2^{k+1}}\right)}\right\}$$

$$\overset{(c)}{\leq} \sum_{k=0}^{\infty} \exp\left(-2^{k-2}\Delta^2\right) \min\left\{1, \frac{2^{k+1}}{n_i}\right\} \overset{(d)}{\leq} \left(\frac{8}{\log(2)} + 8\right) \cdot \frac{1}{n_i \Delta^2} \leq \frac{20}{n_i \Delta^2} \, ,$$

where (a) is just the definition of $Z_i$, (b) follows from the union bound and re-arranging the equation inside the probability, (c) follows from Eq. (3) and the definition of $\log_+$ and (d) is obtained by upper bounding the sum with an integral. $\qquad \square$

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

**A Note on Constants**

The constants in the statement of Theorem 2 can be improved by carefully tuning all thresh-holds, but the proof would grow significantly and I would not expect a corresponding boost in practical performance. In fact, the reverse is true, since the "weak" bounds used in the proof would propagate to the algorithm. Also note that the 4 appearing in the square root of the unbalanced MOSS algorithm is due to the fact that I am not assuming rewards are bounded in $[0, 1]$ for which the variance is at most $1/4$. It is possible to replace the 4 with $2 + \varepsilon$ for any $\varepsilon > 0$ by changing the base in the peeling argument in the proof of Lemma 4 as was done by Bubeck [2010] and others.

**Experimental Results**

I compare MOSS and unbalanced MOSS in two simple simulated examples, both with horizon $n = 5000$. Each data point is an empirical average of $\sim 10^4$ i.i.d. samples, so error bars are too small to see. Code/data is available in the supplementary material. The first experiment has $K = 2$ arms and $B_1 = n^{\frac{1}{3}}$ and $B_2 = n^{\frac{2}{3}}$. I plotted the results for $\mu = (0, -\Delta)$ for varying $\Delta$. As predicted, the new algorithm performs significantly better than MOSS for positive $\Delta$, and significantly worse otherwise (Fig. 1). The second experiment has $K = 10$ arms. This time $B_1 = \sqrt{n}$ and $B_k = (k-1)H\sqrt{n}$ with $H = \sum_{k=1}^{9}1/k$. Results are shown for $\mu_k = \Delta\mathbb{1}\{k = i^*\}$ for $\Delta \in [0, 1/2]$ and $i^* \in \{1, \ldots, 10\}$. Again, the results agree with the theory. The unbalanced algorithm is superior to MOSS for $i^* \in \{1, 2\}$ and inferior otherwise (Fig. 2).

**Figure 1**

**Figure 2:** $\theta = \Delta + (i^* - 1)/2$

Sadly the experiments serve only to highlight the plight of the biased learner, which suffers significantly worse results than its unbiased counterpart for most actions.

## 6   Discussion

I have shown that the cost of favouritism for multi-armed bandit algorithms is rather serious. If an algorithm exhibits a small worst-case regret for a specific action, then the worst-case regret of the remaining actions is necessarily significantly larger than the well-known uniform worst-case bound of $\Omega(\sqrt{Kn})$. This unfortunate result is in stark contrast to the experts setting for which there exist algorithms that suffer constant regret with respect to a single expert at almost no cost for the remainder. Surprisingly, the best achievable (non-uniform) worst-case bounds are determined up to a permutation almost entirely by the value of the smallest worst-case regret.

There are some interesting open questions. Most notably, in the adversarial setting I am not sure if the upper or lower bound is tight (or neither). It would also be nice to know if the constant factors can be determined exactly asymptotically, but so far this has not been done even in the uniform case. For the stochastic setting it is natural to ask if the OC-UCB algorithm can also be modified. Intuitively one would expect this to be possible, but it would require re-working the very long proof.

**Acknowledgements**

I am indebted to the very careful reviewers who made many suggestions for improving this paper. Thank you!

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

# A   Table of Notation

| | |
|---|---|
| $n$ | time horizon |
| $K$ | number of available actions |
| $t$ | time step |
| $k, i$ | actions |
| $\mathcal{B}$ | set of achievable worst-case regrets defined in Eq. (2) |
| $\delta\mathcal{B}$ | boundary of $\mathcal{B}$ |
| $\mu$ | vector of expected rewards $\mu \in [0,1]^K$ |
| $\mu^*$ | expected return of optimal action |
| $\Delta_j$ | $\mu^* - \mu_j$ |
| $\Delta_{ji}$ | $\mu_i - \mu_j$ |
| $\pi$ | bandit strategy |
| $I_t$ | action chosen at time step $t$ |
| $R^\pi_{\mu,k}$ | regret of strategy $\pi$ with respect to the $k$th arm |
| $R^\pi_k$ | worst-case regret of strategy $\pi$ with respect to the $k$th arm |
| $\hat\mu_{k,s}$ | empirical estimate of the return of the $k$ action after $s$ samples |
| $T_k(t)$ | number of times action $k$ has been taken at the end of time step $t$ |
| $i^*$ | optimal action |
| $\log_+(x)$ | maximum of 0 and $\log(x)$ |
| $\mathcal{N}(\mu, \sigma^2)$ | Gaussian with mean $\mu$ and variance $\sigma^2$ |

# B   Proof of Theorem 7

Recall that the proof of UCB depends on showing that

$$\mathbb{E}T_i(n) \in O\left(\frac{1}{\Delta_i^2}\log n\right) .$$

Now unbalanced UCB operates exactly like UCB, but with shifted rewards. Therefore for unbalanced UCB we have

$$\mathbb{E}T_i(n) \in O\left(\frac{1}{\bar\Delta_i^2}\log n\right) ,$$

where

$$\bar\Delta_i \geq \Delta_i + \sqrt{\frac{\log n}{n_i}} - \sqrt{\frac{\log n}{n_{i^*}}} .$$

Define :

$$A = \left\{i : \Delta_i \geq 2\sqrt{\frac{\log n}{n_{i^*}}}\right\}$$

If $i \in A$, then $\Delta_i \leq 2\bar\Delta_i$ and $\bar\Delta_i \geq \sqrt{\frac{\log n}{n_i}}$. Therefore

$$\Delta_i \mathbb{E}T_i(n) \in O\left(\frac{\Delta_i}{\bar\Delta_i^2}\log n\right) \subseteq O\left(\frac{1}{\bar\Delta_i}\log n\right) \subseteq O\left(\sqrt{n_i \log n}\right) \subseteq O\left(\frac{n}{B_i}\sqrt{\log n}\right) .$$

For $i \notin A$ we have $\Delta_i < 2\sqrt{\frac{\log n}{n_{i^*}}}$ thus

$$\mathbb{E}\left[\sum_{i \notin A}\Delta_i T_i(n)\right] \in O\left(n\sqrt{\frac{\log n}{n_{i^*}}}\right) \subseteq O\left(B_{i^*}\sqrt{\log n}\right) .$$

Therefore

$$R^\pi_{\mu,i^*} = \sum_{i=1}^K \Delta_i \mathbb{E}T_i(n) \in O\left(\left(B_{i^*} + \sum_{i\in A}\frac{n}{B_i}\right)\sqrt{\log n}\right) = O\left(B_{i^*}\sqrt{\log n}\right)$$

as required. For the problem-dependent bound we work similarly.

$$R_{\mu,i^*}^\pi = \sum_{i=1}^{K} \Delta_i \mathbb{E} T_i(n)$$

$$\in O\left( \sum_{i \in A} \frac{1}{\Delta_i} \log n + \mathbb{1}\{A \neq \emptyset\} B_{i^*} \sqrt{\log n} \right)$$

$$\in O\left( \sum_{i \in A} \frac{1}{\Delta_i} \log n + \mathbb{1}\{A \neq \emptyset\} B_{i^*} \sqrt{\log n} \right).$$

## C   KL Techniques

Let $\mu_1, \mu_k \in \mathbb{R}^K$ be two bandit environments as defined in the proof of Theorem 1. Here I prove the claim that

$$\mathbb{E}_{\mu_k}^\pi T_k(n) - \mathbb{E}_{\mu_1}^\pi T_k(n) \leq n\varepsilon_k \sqrt{\mathbb{E}_{\mu_1}^\pi T_k(n)}.$$

The result follows along the same lines as the proof of the lower bounds given by Auer et al. [1995]. Let $\{\mathcal{F}_t\}_{t=1}^n$ be a filtration where $\mathcal{F}_t$ contains information about rewards and actions chosen up to time step $t$. So $g_{I_t,t}$ and $\mathbb{1}\{I_t = i\}$ are measurable with respect to $\mathcal{F}_t$. Let $P_1$ and $P_k$ be the measures on $\mathcal{F}$ induced by bandit problems $\mu_1$ and $\mu_k$ respectively. Note that $T_k(n)$ is a $\mathcal{F}_n$-measurable random variable bounded in $[0, n]$. Therefore

$$\mathbb{E}_{\mu_k}^\pi T_k(n) - \mathbb{E}_{\mu_1}^\pi T_k(n) \overset{(a)}{\leq} n \sup_A |P_1(A) - P_2(A)|$$

$$\overset{(b)}{\leq} n\sqrt{\frac{1}{2} \mathrm{KL}(P_1, P_k)},$$

where the supremum in (a) is taken over all measurable sets (this is the total variation distance) and (b) follows from Pinsker's inequality. It remains to compute the KL divergence. Let $P_{1,t}$ and $P_{k,t}$ be the conditional measures on the $t$th reward. By the chain rule for the KL divergence we have

$$\mathrm{KL}(P_1, P_k) = \sum_{t=1}^{n} \mathbb{E}_{P_1} \mathrm{KL}(P_{1,t}, P_{k,t}) \overset{(a)}{=} 2\varepsilon_k^2 \sum_{t=1}^{n} \mathbb{E}_{P_1} \mathbb{1}\{I_t = k\} = 2\varepsilon_k^2 \mathbb{E}_{\mu_1}^\pi T_k(n),$$

where (a) follows by noting that if $I_t \neq k$, then the distribution of the rewards at time step $t$ is the same for both bandit problems $\mu_1$ and $\mu_k$. For $I_t = k$ we have the difference in means is $(\mu_k)_k - (\mu_1)_k = \varepsilon_k$ and since the distributions are Gaussian the KL divergence is $2\varepsilon_k^2$. For Bernoulli random noise the KL divergence is also $\Theta(\varepsilon_k^2)$ provided $(\mu_k)_k \approx (\mu_1)_k \approx 1/2$ and so a similar proof works for this case. See the work by Auer et al. [1995] for an example.

## D   Adversarial Bandits

In the adversarial setting I obtain something similar. First I introduce some new notation. Let $g_{i,t} \in [0, 1]$ be the gain/reward from choosing action $i$ at time step $t$. This is chosen in an arbitrary way by the adversary with $g_{i,t}$ possibly even dependent on the actions of the learner up to time step $t$. The regret difference between the gains obtained by the learner and those of the best action in hindsight.

$$R_g^\pi = \max_{i \in \{1,\dots,K\}} \mathbb{E}\left[ \sum_{t=1}^{n} g_{i,t} - g_{I_t,t} \right].$$

I make the most obvious modification to the Exp3-$\gamma$ algorithm, which is to bias the prior towards the special action and tune the learning rate accordingly. The algorithm accepts as input the prior $\rho \in [0, 1]^K$, which must satisfy $\sum_i \rho_i = 1$, and the learning rate $\eta$.

```
1:  Input: $K, \rho \in [0,1]^K, \eta$
2:  $w_{i,0} = \rho_i$ for each $i$
3:  for $t \in 1, \ldots, n$ do
4:      Let $p_{i,t} = \frac{w_{i,t-1}}{\sum_{i=1}^{K} w_{i,t-1}}$
5:      Choose action $I_t = i$ with probability $p_{i,t}$ and observe gain $g_{I_t,t}$
6:      $\tilde{\ell}_{t,i} = \frac{(1-g_{t,i})\mathbb{1}\{I_t=i\}}{p_{i,t}}$
7:      $w_{i,t} = w_{i,t-1} \exp\left(-\eta \tilde{\ell}_{t,i}\right)$
8:  end for
```

**Algorithm 2:** Exp3-$\gamma$

The following result follows trivially from the standard proof.

**Theorem 8** (Bubeck and Cesa-Bianchi [2012])**.** *Let $\pi$ be the strategy determined by Algorithm 2, then*

$$R_g^\pi \leq \eta K n + \frac{1}{\eta} \log \frac{1}{\rho_{i^*}} .$$

**Corollary 9.** *If $\rho$ is given by*

$$\rho_i = \begin{cases} \exp\left(-\frac{B_1^2}{4Kn}\right) & \text{if } i = 1 \\ (1-\rho_1)/(K-1) & \text{otherwise} \end{cases}$$

*and $\eta = B_1/(2Kn)$, then*

$$R_g^\pi \leq \begin{cases} B_1 & \text{if } i^* = 1 \\ \frac{B_1}{2} + \frac{2Kn}{B_1} \log\left(\frac{4Kn(K-1)}{B_1^2}\right) & \text{otherwise} . \end{cases}$$

*Proof.* The proof follows immediately from Theorem 8 by noting that for $i^* \neq 1$ we have

$$\log \frac{1}{\rho_{i^*}} = \log\left(\frac{K-1}{1 - \exp\left(-\frac{B_1^2}{4Kn}\right)}\right)$$

$$\leq \log\left(\frac{4Kn(K-1)}{B_1^2}\right)$$

as required. $\square$

# E    Concentration

The following straight-forward concentration inequality is presumably well known and the proof of an almost identical result is available by Boucheron et al. [2013], but an exact reference seems hard to find.

**Theorem 10.** *Let $X_1, X_2, \ldots, X_n$ be independent and 1-subgaussian, then*

$$\mathbb{P}\left\{\exists t \leq n : \frac{1}{t}\sum_{s \leq t} X_s \geq \frac{\varepsilon}{t}\right\} \leq \exp\left(-\frac{\varepsilon^2}{2n}\right) .$$

*Proof.* Since $X_i$ is 1-subgaussian, by definition it satisfies

$$(\forall \lambda \in \mathbb{R}) \qquad \mathbb{E}\left[\exp\left(\lambda X_i\right)\right] \leq \exp\left(\lambda^2/2\right) .$$

Now $X_1, X_2, \ldots$ are independent and zero mean, so by convexity of the exponential function $\exp(\lambda \sum_{s=1}^{t} X_s)$ is a sub-martingale. Therefore if $\varepsilon > 0$, then by Doob's maximal inequality

$$\mathbb{P}\left\{ \exists t \leq n : \sum_{s=1}^{t} X_s \geq \varepsilon \right\} = \inf_{\lambda \geq 0} \mathbb{P}\left\{ \exists t \leq n : \exp\left( \lambda \sum_{s=1}^{t} X_s \right) \geq \exp\left( \lambda \varepsilon \right) \right\}$$

$$\leq \inf_{\lambda \geq 0} \exp\left( \frac{\lambda^2 n}{2} - \lambda \varepsilon \right)$$

$$= \exp\left( -\frac{\varepsilon^2}{2n} \right)$$

as required. $\qquad \square$