[Reviews · NeurIPS 2015]

Submitted by Assigned_Reviewer_1

(As requested by NIPS, this is a "light" review)

SUMMARY

The papers considers a stochastic bandit problem and analyzes a variant of minimax regret: Instead of studying the regret against the worst set of distributions, they consider for each arm k\in[K] the regret against the worst set of

distributions under the constraint that this arm is optimal. This leads to a richer notion of minimax-regret, seen as a vector in R^K.

The paper studies this notion of minimax regret and provide both lower bounds and upper bounds, expressed in terms of a Pareto-regret frontier. The proof for the lower bound is specific and simple enough. The algorithm for the upper bound, called Unbalanced MOSS,

is a generalization of the MOSS algorithm.

Some illustrative examples are provided. They also show that the additional factor K in the regret of MOSS is not just an artifact of the original proof and is observed in practice. They then suggest an unbalanced UCB algorithm.

QUESTIONS AND COMMENTS

The paper restricts to rewards distributed according to a normal distributions. This yields some questions: 1) To which extent your proofs are married with this assumption? What cannot be easily extended? 2) What would be the main modifications for the traditional [0,1] bounded rewards?

3) What about simply dealing with a sub-Gaussian/light-tail assumption?

I believe section 4 could be detailed more, and enriched with further discussions. Page 8 contains many additional remarks that are however not discussed at all. They seem to have been assembled in a rush. Theorem 7 and especially the paragraph on Adversarial Bandits should be discussed more, especially against the state-of-the-art.

Figure on page 8 is hard to read when printed in B&W.

As it is, the paragraph on Adversarial Bandit looks misplaced/incoherent with the paper: I suggest to either discuss it at length, maybe in a dedicated section, or skip it.

You introduced/studied Unbalanced-MOSS and Unbalanced-UCB : What about Unbalanced-KL-UCB or Unbalanced-Thompson sampling? (It would make sense to study these at least empirically).

Quality: Looks ok, I haven't checked in detail since this is a "light" review. Clarity: Ok. The paper seems messy and should be polished/reorganized. More intuition/discussion should be provided. Originality: Good. Significance: The paper seems to bring an interesting/useful idea. The results should be discussed more against the state-of-the-art and the literature review part is a bit weak.

* I have read the rebuttal and keep my score unchanged.
Summary: The paper seems to bring an interesting/useful idea. The writing is a bit messy and

the paper should be polished/reorganized. More intuition/discussion should be provided as well. The results should be discussed more against the state-of-the-art and the literature review part is a bit weak.

Submitted by Assigned_Reviewer_2

Summary: In a multi-armed bandit problem, define the worst-case regret of a strategy for a given action as the largest possible regret incurred when the given action is an optimal one. Further define the worst-case regret vector of a strategy as the vector of worst-case regrets for different actions. The paper fully characterizes, up to constant factors, the set of achievable worst-case regret vectors in the stochastic setting. These regret vectors can be achieved by a variant of the MOSS algorithm by Audibert and Bubeck [2009]. The authors also showed empirically that MOSS does not achieve near-optimal problem-dependent guarantees. Moreover, in the adversarial setting, they proved that a variant of the EXP strategy can lead to a certain family of worst-case regret vectors.

Comments:

Given that the problem considered in this paper is a natural and seemly simple extension of the classic multi-armed bandit problem, it is a little surprising that it has not yet been solved before. The results in the paper are novel and important. I like the paper for three reasons.

First, it is important to quantify the cost of falsely believing that a certain action is optimal and as a result, choosing that action more frequently than its past performance suggests. Second, while the KL-divergence argument is usually used to get a lower bound on the sum of regrets, in this paper, it is applied in a clever way to lower bound the product of regrets. finally, it is interesting to see that a simple but clever modification of the UCB and MOSS algorithms can match the lower bounds.

The paper is well organized and its contribution is clearly stated. However, the proofs in the paper contain many confusing errors and typos, although I think that they could be corrected without modifying the structures of the proofs.

Proof of Theorem 1: - line 140:

-\epsilon_1 should be -\epsilon_j. Otherwise, the second equality in line 157 would be wrong.

- line 157: one of the two T_k's should be T_1.

Proof of Theorem 7 in the appendix - line 485: the "-" should be a "+" and the "+" should be a "-". - line 516: B_1 in the third term of the bound should be

B_{i^*}. - line 516-518: the inequality \sum_{i=2}^{K} \frac{n}{B_i} \leq B_1 is used. However, if 1 is in the set A_1, then there would be an extra term \frac{2n}{B_1}\sqrt{log(n)} that need to be dealt with.

- line 518: the desired bound should be {B_{i*}}\sqrt{log(n)} instead of {B_1}\sqrt{log(n)}.

To summary, the paper solves an interesting problem by cleverly adapting existing methods and algorithms. It provides some valuable insights into the achievable worst-case regrets in the multi-armed bandit problem. However, the proofs in the paper contain some errors. Although I think that the results of the paper are still correct, the paper is not ready to be published unless the errors can fixed.

After rebuttal:

The author rebuttal has answered my comcern. I think that the paper studies a meaningful and important problem thoroughly by providing matching lower and upper bounds. I raise my score to 7.
Summary: The paper solves an interesting problem by cleverly adapting existing methods and algorithms. It provides some valuable insights into the achievable worst-case regrets in the multi-armed bandit problem. However, the proofs in the paper contain some errors. Although I think that the results of the paper are still correct, the paper is not ready to be published unless the errors can fixed.

After rebuttal: The author rebuttal has answered my concern. I raise my score to 7 and vote for acceptance.

Submitted by Assigned_Reviewer_3

This paper addresses the minimax bandit problem in the stochastic setting. The case that is targeted is to bound specifically the regret for a particular arm and characterised how this deteriorates the regret on the remaining arms. The authors provide new theoretical material to prove an upper and lower bound on the repartition of the per-arm regret that match up to constants. The upper bound is based on the proposed

new algorithm Unbalanced-Moss which is based on the Moss algorithm. They show that unlike in the expert case, guaranteeing a small regret for a particular arm comes to a greater cost (than in the expert case) for the remaining arms. Finally they design a new version of UCB (unbalanced UCB), that up to a logarithmic factor is 'unfair minimax' optimal and which at the same time and contrary to the MOSS algorithm possess asymptotic optimal problem-dependent regret.

The paper addresses a problem of importance and seems in the most part of high value. My main concern is that I am not sure about the correctness of the lower bound proof. The question raised there should definitely be addressed.

Some part of the proofs are quite compact and makes them hard to verify: passage (c)&(d) of proof of theorem 3. More intuition should be given to ease the reading of the proofs.

--About the lower bound:

Equation in line 143: Shouldn't it be an inequality instead of the last equality, as \epsilon_k is defined with respect to R_k which is a sup on R_\mu?

In line 126, it is assumed that \mu \in

[0,1]**K, but in the example given in the proof (the \mu_k) it seems not always the case. I think this is an issue because, as mentioned above about line 143, we need to use

that R_k\geq R_\mu_k. You can see that \mu_k do not belong to [0,1]**K first of all because there are negative numbers (-\epsilon_1). Second, isn't there an issue with the constant c? In the end the constant c is put to 4 but it seems that, in some cases, the constant should be smaller than 1 by construction. Indeed we want \epsilon_k \leq 1 so that the \mu_k belong to [0,1]**K. So if the strategy \pi is such that the regret R_k on arm k is linear (R_k=n), as it could happen if the strategy never pulls k, then we would have \epsilon_k=c. So c\leq 1. But this creates a problem with the end of the proof where we need c > 2.

In the passage (a) of line 150 why is there a T_k(n) appearing in the square root? It is not obvious from the mentioned standard entropy inequality.

equation line 157: on the first equality the \epsilon_k should be \epsilon_1 as defined in \mu_1. This seems to create problem for the following reasoning in the same line. In this same line, in the 'last but one' term, I guess it should be already R_k instead of R_\mu_k

More generally, shouldn't the constant c depend on \pi? Because otherwise the Equation 2 is proving that any possible strategy is pulling all the arms at least linearly! (when c=4, as used in the proof).

-- Other possible typos in the rest of paper: line 269: In the second part, at the end of the line: where is it proven that \Delta_i\E\tau_i\leq \sqrt{n_i*}. Is it useful? I see the part with n_i but not with n_i*.

line 289: a should be \alpha?

line (d) of Corollary 6. Shouldn't it be \alpha+1 and \alpha+1/2 in the denominator? Then the 'proof by graph' becomes even more fishy if their is a typo in the expression (though I guess it is not changing much).

line 305: (b) it should be \Delta_i\geq 4*sqrt(n_i*) right?

line 311: \tilde \Delta_l is not defined for l=0.
Summary: The paper proposes new original theoretical techniques and results that address the minimax problem in stochastic bandit. The results are of importance and the work seems of high value. However I have an issue with

the proof of the lower bound that needs to be solved for a possible publication. I am willing to change my score upon reading the authors rebuttal.

Submitted by Assigned_Reviewer_4

The paper deals with the problem of unfair bandits, where we would like to force different minimax regret values for the different arms. The motivation comes from cases where we have prior information that some arms are more likely to be optimal compared to other arms. This problem has been previously studied in the experts setting where the player has access to all of the rewards. The authors provide a lower bound for the problem, along with a matching upper bound ("unbalanced MOSS") for the stochastic case, thus giving a complete, up to constant factors, characterization of the attainable minimax bounds. The characterization is somewhat surprising and shows that the bandit case is much more strict than the expert case, in the sense that forcing a small regret in one arm comes with a heavy price w.r.t the regret of the other arms. The authors further provide a modification of UCB that achieves problem dependent bound, and a modification of EXP3 for the adversarial case.

The problem being dealt with is original and interesting, and definitely deserves an analysis. The techniques used are not exceptionally novel but are clean, elegant, and get the job done. Overall the paper is well written: The motivation is properly explained, the previous works are properly surveyed as far as I could verify, the proofs are overall well explained, well structured and have a nice flow to them.

There is one issue with the proofs that I was unable to verify and would like the authors to explain in their response, and fix in the camera-ready version, if accepted. The score I gave the paper is pending on a satisfactory answer. Other than this hiccup I find it to be a solid paper: In page 3, line 150, the inequality marked by (a): This inequality is non-trivial and the main theorem is based on it. It deserves a much more detailed explanation than the one given in the footnote. In particular, please state exactly how P and Q are defined (are they the distributions associated with \mu_1 and \mu_k)? Also, state how is X defined and how does it relate to P and Q.

Finally and most importantly, please provide a formal statement of the used fact, along with a proper citation.

Minor comments / typos: Page 2, definition of {\cal B}: All B_i's should be positive Page 2, line 78: Mention that B_K is the largest value of B Page 3, line 140: Should the "otherwise" option be -\eps_j rather than -\eps_1 ? Page 4, line 174: B_1 \geq \Sum ... => B_1 \geq (1/8)* \Sum ...

Page 4, just before lemma 3: A suggestion: I would add a 1-liner explaining what lemmas 3 and 4 are about: lemma 3 is to lower bound the estimation of \mu_1 and lemma 4 to realize how soon we have a proper upper bound of each mu_i

Page 5, line 222: 2^{k} \Delta => 2^{k+1} \Delta Page 5, line 256: Should be: \forall s' \leq s, \hat{\mu}_{i,s'} - \mu_{i,s'} <=... Page 6, line 305, equation after (b): 2/\Delta -> 4/\Delta Page 6, line 310: The verbal description of A is wrong. It contains the arms that are not negligibly sub-optimal
Summary: The paper deals with the problem of unfair bandits, where we would like to force different minimax regret values for the different arms. The problem is motivated and has been studied in the expert case. The paper provides a thorough and interesting analysis of the problem, giving lower bounds and an algorithm for a matching upper bound. It has some issues but is overall well written, and interesting.

Author Feedback
Author rebuttal: All Rev:

Thank you to all reviewers for carefully reading this submission.

All reviewers noted typos and minor errors and we are sorry for inflicting those upon you, and thankful for your perseverance. Fortunately all the errors are relatively minor and are solved either by fixing the typos or more carefully stating the required assumptions.

We feel this paper can be made acceptable without much difficulty and hope that it can get over the bar for NIPS. Besides polishing, our plan for improvement is:

1. Fix minor errors/typos and correct the boundedness assumption and restate the upper bound.
2. Add intuition and extend the lit review. This will come at the cost of pushing some lemmas
to the appendix (in hindsight this is worthwhile)

See specific responses below.

Rev. 1:

For Thm 1 you are right. -epsilon_1 should be -epsilon_j. The other point is right too

For Thm 7, your 2nd last comment seems not a problem, since i^* is never in A_1 or A_2, but we should point this out

Rev. 2:

>...line 143: Shouldn't it be an inequality
Yes
>...assumed that \mu \in [0,1]**K
Yes, this is an issue. The simplest solution is to drop the assumption of bounded rewards and restate the upper bound. This
makes no difference in the interesting regime where R_k^n / n <= 1 for all k
>...T_k(n) appearing in the square root?
See response to reviewer 3
>...\epsilon_k should be \epsilon_1 as defined in \mu_1
There is a typo in the def. of mu_k, where -eps_1 should be replaced by -eps_j
>...R_k instead of R_\mu_k
Yes
>...shouldn't the constant c depend on \pi?
No. (2) is about the number of pulls of arm k in environment k
>line 269: In the second part...
Should be dropped
>line 289: a should be \alpha?
Yes
>Shouldn't it be \alpha+1
Yes. Upper bound becomes 8 / \sqrt(e) by simple analysis
>should be \Delta_i\geq 4*sqrt(n_i*)
Yes

Rev. 3

We will incorporate your suggestions

Re. line 150: X = T_k(n) satisfies |X| <= n. P is a measure on the space of outcomes given environment \mu_1 and Q
is a measure on the space of outcomes given environment \mu_k (both depend on choices of the fixed policy).
Assume WLOG that k = 2

\mu_1 = (0, -eps_2, -eps_3,..., -eps_K)
\mu_2 = (0, eps_2, -eps_3,...,-eps_K)

The two differ in the 2nd term. Def. P_t and Q_t as conditional measures on the reward at time step t given the history, including
arm chosen at time step t

By the chain rule: KL(P,Q) = sum_{t=1}^n E_P[KL(P_t, Q_t)].

If the arm at time step t is not 2, then KL(P_t, Q_t) = 0. Else it is the divergence between
two Gaussians with variance=1 and means differing by 2eps, which is 2eps_2^2.

Therefore E_P[KL(P_t,Q_t)] = E_P[indicator(I_t=2) 2eps_2^2] and so

sum_{t=1}^n E_P[KL(P_t,Q_t)] = 2eps_2^2 E_P T_2(n)

We will add this and reference [1] where it is described in detail.

Rev. 4:

We will proof-read the paper for other such ambiguities.

Rev. 5:

We know the nice paper by Bubeck&Liu, but don't see the relevance. They bound the Bayes regret in a prior free way and
otherwise focus on the BPR setting where specific priors are crafted for good frequentist regret with Thompson sampling. They do
not analyse the general case or the effects of the choice of prior. These results have no implications on the available trade-offs to Bayesian algorithms.

There is an arxiv paper by Liu&Li (unavailable at submission time). They
study a similar problem with two arms and focus on Thompson sampling. We will include a reference/discussion to this paper.

We agree a Bayesian approach is natural, but one needs to understand the effects of the prior on the regret. We
characterises what is possible for *any* algorithm. Problem-dependent regret is important, but we find the logarithmic regime
less interesting here because it does not match the motivation to design algorithms that come with solid guarantees on the regret
regardless of the specific problem. The Liu&Li paper also focusses on the worst-case.

We extended MOSS to obtain matching bounds. Generalising other variants of UCB is quite easy.

See response to reviewer 2 for issue with bounded means.

We hope to see an increase in your score.

Rev. 6:

>...married with this assumption?
This is for the lower bound only, which hold for most natural noise models, like Bernoulli.

>...modifications for the traditional [0,1]...
For the lower bound you need to compute and bound the KL divergence for Bernoulli rewards, which is done in [1]
and would not change the nature of the bounds.

>...sub-Gaussian...
Lower bound doesn't hold for distributions much more concentrated than a gaussian. Upper bounds hold essentially unchanged.

The proof for unbalanced UCB is general and can be carried through for any algorithm with
suitable bounds on E T_k(n), including KL-UCB/UCB-V and possibly Thompson sampling.

[1] Auer et. al. Gambling in a rigged casino: The adversarial multi-armed bandit problem